# Descriptive study of chest x-ray examination in mandatory annual health examinations at the workplace in Japan

Yuya Watanabe[1], Toru Nakagawa[1], Kota Fukai[2], Toru Honda[1], Hiroyuki Furuya[2], Takeshi Hayashi[1], Masayuki Tatemichi[2]*

1 Hitachi Health Care Center, Ibaraki, Japan, 2 Department of Preventive Medicine, Tokai University, School of Medicine, Kanagawa, Japan

* tatemichi@tokai-u.jp

**Data Availability Statement:** Most all relevant data are within the paper and its Supporting Information files. For raw data, study participant institutions did not consent to have their data publicly available and

## Abstract

The utility of chest x-ray examination (CXR) in mandatory annual health examinations for occupational health is debatable in Japan. This study aimed to provide basic data to consider future policies for mandatory annual health examinations in the workplace. A nationwide descriptive survey was performed to determine the rate of detection of tuberculosis, lung cancer, and other diseases through CXR in organizations associated with National Federation of Industrial Health Association. The rate of finding on CXR conducted during annual health examinations in FY2016 was evaluated. Data regarding diagnosis based on follow-up examination findings were obtained and compared with the national statistics. In addition, CXR findings were compared with the results of low-dose lung computed tomography performed at the Hitachi Health Care Center. From 121 surveyed institutions, 88 institutions with 8,669,403 workers were included. For all ages, 1.0% of examinees required follow-up examination. Among 4,764,985 workers with diagnosis data, the tuberculosis detection rate was 1.8–5.3 per 100,000 persons. For Lung cancer, 3,688,396 workers were surveyed, and 334 positive cases were detected. The lung cancer detection rate using CXR was 9.1–24.4 per 100,000 persons. From 164 cases with information regarding the clinical stage, 72 (43.9%) had Stage I lung cancer. From 40,045 workers who underwent low-dose computed tomography multiple times, 31 lung cancer cases, all with Stage I disease, were detected (detection rate: 77.4 per 100,000 persons). Our findings suggest that CXR plays a little role in the detection of active tuberculosis. With regard to LC screening, the detection rate of LC by CXR was lower, approximately 50%, than the expected rate (41.0 per 100,000 persons) of LC morbidity based on the age–sex distribution of this study population. However, the role of CXR for LC screening cannot be mentioned based on this result, because assessment of mortality reduction is essential to evaluate the role.

## Introduction

Periodic chest x-ray examination (CXR) in occupational settings has been conducted for active tuberculosis (TB) screening since 1972, as required by the Industrial Safety and Health Law

freely accessible, and we are therefore unable to share the data publicly online for ethical reasons. However, the raw data may be available after review of the purpose and permission by the study committee. Requests must be sent to the Tokai University School of Medicine Clinical Research Review Board (tokai-rinsho@ml.tokai-u.jp).

**Funding:** This study was supported by the Ministry of Health, Labour, and Welfare of Japan through an Industrial Disease Clinical Research Grants (grant no. 170301). The Hitachi Health Care Center employs Y.W., T.N., T.Ho., and T.Ha. for conducting medical examinations, but did not have any additional role in the study design, data collection and analysis, decision to publish, or preparation of the manuscript. The specific roles of these authors are articulated in the 'author contributions' section.

**Competing interests:** The all authors have declared that no competing interests exist. Y.W., T.N., T.Ho., and T.Ha. are employees of Hitachi Health Care Center. This does not alter our adherence to PLOS ONE policies on sharing data and materials.

[1], subsequent to the TB Prevention Law of 1950 in Japan. The usefulness of CXR in mandatory annual health examinations has been vigorously discussed; however, there is no consensus on whether it should be a mandatory examination for all workers aged ≥40 years [2]. In particular, as the incidence of TB has declined, the significance of CXR TB screening has been increasingly debated [2]. In recent years, the digitalization of chest photographs has been promoted instead of mass miniature radiography (MMR), and the detection rate is expected to be consequently improved [3].

It is undeniable that the primary purpose of CXR in the workplace is to detect workers with active TB [1]. In 2016, the prevalence of TB had decreased to 13.9 per 100,000 people in Japan [4]. However, in 2015, with the aging of the working population, the prevalence of lung cancer (LC) had increased to 8.8–278.5 in men and 6.9–111.3 in women in the age group of 40–69 years [5]. The usefulness of any examinations for screening depends on the pre-test probability and accuracy of findings [6]. Therefore, the clinical utility of CXR should be discussed as relevant in each era.

In addition to these two major diseases, much information can be obtained from CXR in the clinical situation. CXR can detect not only occupational lung diseases, such as pneumoconiosis, obstructive pulmonary disease, pneumonia, mediastinal tumor, pneumothorax, and pleural effusion, but also atherosclerotic lesions, cardiac enlargement, aortic aneurysms, and so on. However, it is necessary to evaluate the significance of the annual CXR from various perspectives. Therefore, there has been much debate on the need for CXR in the mandatory health examinations at the workplace [7,8].

Japanese guideline for lung cancer screening recommends CXR [9]. Thus mandatory CXR in workplace plays a role in lung cancer screening in addition to screening for TB. The first priority of evaluation of effectiveness on cancer screening must be based on randomized controlled trials (RCT) using mortality as outcome. Theoretically the detection number of cancers is not useful outcome to evaluate effectiveness, because cancers detected by screening can include cases with overdiagnosis. Thus this study did not aim to evaluate the effectiveness of CXR for cancer screening. The purpose of this study is to provide actual descriptive data on the current state of social implementation of the screening as a measure, not directly to assess the effectiveness of CXR for the cancer screening.

To provide basic data of mandatory annual workplace CXR screening for workplace health, we conducted a nationwide descriptive survey to determine the detection rate of TB, LC, and other diseases through CXR.

## Participants and methods

### CXR

National Federation of Industrial Health Organization (NFIHO) contracted with 121 occupational health organizations or institutions (across 43 of the 47 prefectures in Japan) that were eligible for collection of the CXR results. NFIHO comprises the largest group of occupational health organizations that conduct legally mandated health examinations, provide health guidance, and issue occupational health improvement guidance based on the Japanese Industrial Health and Safety Law. Moreover, NFIHO plays an important role in ensuring quality control of medical tests for over 46 million of the 56 million workers in Japan, including the annual mandatory health examinations of 14 million individuals.

From the occupational health organizations or institutions associated with NFIHO that responded positively to our request, we obtained data regarding the actual number of CXRs performed (regardless of whether they were digital or analog x-rays), the number of findings, and the number of examinees who required follow-up examinations during mandatory health

examinations, including comprehensive workplace health check-up, for the period from April 2016 to March 2017. Data were collected for all examined workers and for workers aged ≥40 years. The number of examinees who required further follow-up examinations and the number of examinees suspected to have TB were investigated, and the rates for each parameter were calculated.

In addition, when examinees could be followed up and had a definitive diagnosis, such as TB, LC, or other diseases, additional information was obtained. The rates of detection for TB and LC were calculated from the number of examinees who were diagnosed with active TB and LC, respectively. Concerning LC, data regarding the histopathological type and clinical stage were recorded. Furthermore, details of diseases other than TB and LC that necessitated further follow-up examination were extracted.

## Low-dose computed tomography (LDCT) screening for LC

Detailed information on LDCT screening has been described in previous reports [9–11]. Briefly, since April 1998, LDCT has been performed at the Hitachi Health Care Center for individuals aged ≥50 years. From April 1998 to December 2006, the imaging was performed on a single-row CT (conditions: 120 kV, 50 mA, 10 mm collimation, pitch 2). From January 2007, a new model with 4-row multi-detector row CT was used (3.75 mm × 4 rows, 120 kV, 20 mA, 5-mm collimation, pitch 5, computed tomography dose index: 1.4 mGy). From the beginning of 1998, the system was implemented using a comparative reading system and a double reading system. In cases of examinees who required follow-up examinations, further investigations were conducted at the Hitachi General Hospital. In the present study, we collected the results for 10 years from April 1998 to March 2009. We subdivided the examinees into groups of those who underwent one CT screening and those who underwent two or more CT examinations, and tabulated the number of detailed examinations. Moreover, data regarding the cancer pathology and clinical stage were obtained, and the results from 2016 to 2018 were aggregated to evaluate the latest rate of detection.

## Information on death due to LC

Details of workers who died of LC from April 2010 to March 2018 were retrospectively examined with regard to information on the history of LDCT, histological types, and clinical stage. These abovementioned data were obtained from the death certificate of the worker, and the worker's examination history of LDCT was traced back up to 2 years and details of smoking status were also obtained.

This study was approved by the Tokai University School of Medicine Clinical Research Review Board (approval no. 16R-076, 19R-228). In the survey, information on the number of CXRs and LDCTs that were conducted and the number of findings was collected. None of the abovementioned data contained personal information, and there was no possibility of violation of patient confidentiality or personal information being accessed. Thus, the need for informed consent was waived.

## Statistical analysis

For CXR, the rate of follow-up examinations and the rate of detection of TB and LC were calculated from the number of examinees that required follow-up examination, were suspected to have TB, were diagnosed with TB or LC as a numerator, and undertook CXR as a denominator.

The rate obtained by dividing the number of examinees who actually underwent follow-up examinations by that of examinees who required follow-up examination was defined as the

rate of follow-up examination. The estimated rate of detection was calculated by dividing the actual rate of detection by the rate of follow-up examination. These data were expressed as the number of findings per 100,000 persons. The age–sex distribution over 5 years for examinees who underwent CXR in occupational health organizations or institutions associated with NFIHO were separately collected. The expected incidence rate was calculated from the reports of the Japanese government concerning TB [4] or cancer statistics database [6]. The 95% confidence interval (CI) was determined using the Agresti–Coull method.

## Results

With the cooperation of 88 facilities (72.7%) from 121 affiliated members of NFIHO, we included data from a total of 8,594,676 workers (men: 5,461,011, women: 3,133,665) in this study. The prevailing rate of analog CXR was 18.9%. The rates of workers who were aged ≥40 years were 57.7% and 59.5% in men and women, respectively.

The results of CXR are shown in Table 1. Of the 5,461,011 men examined, the rate of abnormal findings was 9.1% (498,351) and that of examinees who required re-examination was 0.6%. The rate of examinees who required further follow-up examinations was 1.1%, and the suspicion rate for TB was 0.0068%. Among the 3,133,665 women, the rate of abnormal findings was 6.7%, and the rate of examinees who required re-examination was 0.49%. The rate of examinees who required further follow-up examinations was 0.9%, and the suspicion rate for TB was 0.0056%.

## Detection of tuberculosis

Forty-four institutions had follow-up data on the precise diagnosis, and data from 4,764,985 individuals (2,991,434 men, 1,773,551 women) of all ages were available for the calculation of the TB detection rate. Table 2 shows the results of the calculations. Of the 10,460 men and 6,174 women of all ages who underwent follow-up examinations and were diagnosed, 88 cases (59 men, 29 women) were determined to have active TB. The positive predictive value (PPV) was primarily calculated to be 0.18% (88/47,962). This value might have been an underestimation, because the cases that could be followed up were 16,734 (34.9%). Thus, the maximum PPV was calculated as 0.56% (88/16734), which might have been an overestimation. Plausible PPV was estimated to range between 0.18% and 0.56%. Overall, the detection rates of TB were 2.0 and 1.5 per 100,000 persons in men and women, respectively. In total, 88 cases were detected in this study, suggesting that the detection rate was 1.8 (95% CI: 1.5–2.2) per 100,000 persons. As the rate of examinees who could be followed up was 32.8% in men and 39.0% in

**Table 1. Results of chest x-ray examination.**

| | All age groups | | | ≥40 years | | |
|---|---|---|---|---|---|---|
| | **Men** | **Women** | **Total** | **Men** | **Women** | **Total** |
| Examinees | 5,461,011 | 3,133,665 | 8,594,676 | 3,192,508 | 1,854,750 | 5,047,258 |
| Number of participants with abnormal findings | 498,351 | 210,339 | 708,690 | 376,010 | 159,435 | 535,445 |
| Rate of abnormal findings | 9.1% | 6.7% | 8.2% | 11.8% | 8.6% | 10.6% |
| Number of participants who required follow-up examinations | 58,130 | 27,016 | 85,146 | 48490 | 25912 | 74402 |
| Rate of participants who required follow-up examination | 1.10% | 0.86% | 0.99% | 1.50% | 1.4% | 1.47% |
| Number of participants who had findings suspicious for tuberculosis | 374 | 176 | 550 | 317 | 125 | 442 |
| Rate of participants who had findings suspicious for tuberculosis (per 100,000 persons) | 6.8 | 5.6 | 6.4 | 9.9 | 6.7 | 8.8 |
| 95% confidence interval | 6.2–7.5 | 4.8–6.5 | 5.8–6.9 | 8.8–11.0 | 5.5–7.9 | 7.9–9.6 |

**Table 2. Detection of tuberculosis and other diseases using chest x-ray examination among participants who were followed up for information on detailed examination.**

| | All ages | | | ≥40 years | | |
|---|---|---|---|---|---|---|
| | **Men** | **Women** | **Total** | **Men** | **Women** | **Total** |
| Examinee | 2,991,434 | 1,773,551 | 4,764,985 | 1,757,142 | 1,048,646 | 2,805,788 |
| Number of participants who required follow-up examination | 31,891 | 16,071 | 47,962 | 26,339 | 16,827 | 43,166 |
| Rate of follow-up examination requirement | 1.07% | 0.91% | 1.0% | 1.50% | 1.60% | 1.54% |
| Number of participants who underwent follow-up examination | 10,460 | 6,274 | 16,734 | 7,309 | 4,880 | 12,189 |
| Rate of follow-up examination | 32.8% | 39.0% | 34.9% | 27.7% | 29.0% | 28.2% |
| Number of cases diagnosed with tuberculosis | 59 | 29 | 88 | 36 | 17 | 53 |
| Positive predictive value for tuberculosis screening Actual PPV (maximum) PPV)* | 0.19% (0.56%) | 0.18% (0.46%) | 0.18% (0.53%) | 0.14% (0.49%) | 0.14% (0.35%) | 0.12% (0.43%) |
| Rate of detection of tuberculosis (per 100,000 persons) | 2.0 | 1.6 | 1.8 | 2.0 | 1.6 | 1.9 |
| 95% confidence interval | 1.5–2.5 | 1.0–2.2 | 1.5–2.2 | 1.4–2.7 | 0.8–2.4 | 1.4–2.4 |
| Estimated rate of tuberculosis (per 100,000 persons)** | 6.0 | 4.2 | 5.3 | 7.4 | 5.6 | 6.7 |
| 95% confidence interval | 5.1–6.9 | 3.2–5.2 | 4.6–5.9 | 6.1–8.7 | 4.1–7.0 | 5.7–7.7 |
| Number of participants with x-ray findings of other diseases | 3,246 | 2,050 | 5,296 | 2,719 | 1,825 | 4,544 |
| Rate of other diseases | 0.11% | 0.12% | 0.11% | 0.15% | 0.17% | 0.16% |

*Maximum PPV was calculated by the actual PPV divided by the rate of follow-up examination.

** The estimated rate of tuberculosis was calculated by the number of cases diagnosed with tuberculosis divided by the rate of follow-up examination.

PPV, positive predictive value.

women, the detection rate was approximately estimated to be 5.3 (95% CI: 4.6–5.9) per 100,000 persons (calculated as follows: [59/0.328+29/0.90]/4,764,985).

According to the national database of TB in 2016 [4], the incidence rate of TB was 13.9 per 100,000 persons. Moreover, the rate of new TB patients among workers was 57.0% [4], and the estimated incidence rate among workers was 13.9×0.57 = 7.9 per 100,000 persons. According to the stratification of the TB incidence rate by age–sex in 2016 [4], the expected incidence of TB was calculated to be 9.1 (95% CI: 8.2–9.9) per 100,000 persons based on the age–sex distribution of NFIHO (S1 Table).

## Detection of LC

With regard to LC, 38 institutes had follow-up data on precise diagnoses. Moreover, data from 3,688,365 individuals (2,295,702 men, 1,392,693 women) of all ages were available (Table 3). Of 25,442 men and 12,704 women who were screened and found to require follow-up examination for LC, 8,849 men and 5,288 women underwent detailed examinations. The rates of institutional follow-up examinations were 34.8% and 41.6% in men and women, respectively.

Subsequently, 334 cases were diagnosed as LC, and the detection rate was 9.1 (95%CI: 8.1–10.0) per 100,000 persons. Overall, the total number of participants diagnosed with LC was estimated to be 672 (234/0.348) in men and 240 (100/0.416) in women, and a total incidence rate of 24.7 (95% CI: 22.8–26.0, 912/3,688,395) per 100,000 persons was noted. According to the LC incidence rate in 2015 stratified by age–sex [4], based on the age–sex distribution of NFIHO data (S1 Table), the expected LC incidence rate was 41.0 (95% CI: 39.2–42.9) per 100,000 persons.

Of the 344 LC cases, pathological and clinical stage information was obtained from 164 cases; 72 (43.9%) had Stage I disease, 21 (12.8%) had Stage II, and 32 (19.5%) had Stages III or IV. With regard to pathology, 123 (74.5%) cases were adenocarcinomas, 29 (17.6%) were

**Table 3. Detection of lung cancer using chest x-ray examination among participants who were followed up for information on detailed examination.**

| | All ages | | | ≥40 years | | |
|---|---|---|---|---|---|---|
| | **Men** | **Women** | **Total** | **Men** | **Women** | **Total** |
| Examinee | 2,295,702 | 1,392,693 | 3,688,395 | 1,385,574 | 832,901 | 2,218,475 |
| Number of participants who required follow-up examination | 25,442 | 12,704 | 38,146 | 21,208 | 14,026 | 35,234 |
| Rate of follow-up examination requirement | 1.11% | 0.91% | 1.0% | 1.53% | 1.68% | 1.59% |
| Number of participants who underwent follow-up examination | 8,849 | 5,288 | 14,137 | 6,027 | 4,077 | 10,104 |
| Rate of follow-up examination | 34.8% | 41.6% | 37.1% | 28.4% | 29.1% | 28.7% |
| Number of cases diagnosed with lung cancer | 234 | 100 | 334 | 228 | 98 | 326 |
| Positive predictive value for lung cancer screening Actual PPV (maximum PPV)* | 0.92% (2.64%) | 0.79% (1.89%) | 0.88% (2.39%) | 1.08% (3.78%) | 0.70% (2.40%) | 0.93% (3.23%) |
| Rate of detection of lung cancer (per 100,000 persons) | 10.2 | 7.2 | 9.1 | 16.5 | 11.8 | 14.7 |
| 95% confidence interval | 9.1–11.8 | 5.7–8.6 | 8.2–10.2 | 14.3–18.6 | 9.4–14.1 | 13.1–16.3 |
| Estimated rate of lung cancer (per 100,000 persons)** | 29.3 | 17.3 | 24.4 | 57.9 | 40.5 | 51.2 |
| 95% confidence interval | 27.8–32.3 | 15.1–19.4 | 23.2–26.5 | 53.9–61.9 | 36.1–44.8 | 48.3–54.2 |

* Maximum PPV was calculated by the actual PPV divided by the rate of follow-up examination.

**The estimated rate of lung cancer was calculated by the number of cases diagnosed with lung cancer divided by the rate of follow-up examination.

PPV, positive predictive value.

squamous cell carcinomas, five (3.0%) were small cell carcinomas, and one (0.6%) was a large cell carcinoma (Table 4).

## LDCT screening

Table 4 shows the results of LDCT. The first examination detected 60 cases with LC, and the rate was 386 (95% CI: 287.4–485.6) per 100,000 persons. Second or later examinations revealed that the detection rate was 77.4 (95% CI: 49.3–105.5) per 100,000 persons, and 31 cases (100%) exhibited clinical Stage I disease. Fig 1 shows the association between the numbers of times that examinees underwent LDCT and the rate of detection, which decreased in proportion to the number of times LDCT was conducted.

S2 Table shows the recent (2016–2018) number of patients who underwent LDCT on stratification by age–sex, the number of required follow-up examinations, and number of LC. Of the 11,632 (10,257 men and 1,375 women) patients, three cases with LC were detected, and the incidence rate was 25.8 (95% CI: 00.0–63.1) per 100,000 persons. Based on the age–sex distribution of recent Hitachi data, the expected LC incidence rate was 155.2 (95% CI: 80.3–230.9) per 100,000 persons.

## Death due to LC

During the follow-up period from 2010 to 2018, 17 workers died because of LC. The pathologic diagnosis, clinical stage, and history of LDCT screening are shown in S3 Table. Of the seven workers who died of small cell carcinoma, three (42.9%) men had a history of LDCT screening. In addition, all of them had a history of current or past smoking. Of the five patients who died because of adenocarcinoma, one (20%) had a history of LDCT screening.

## Detection of other diseases

S4 Table lists the findings, other than TB and LC, which were identified in this study among 4,764,985 workers (2,991,434 men, 1,773,551 women). The numbers of cases of patients with emphysema, nontuberculous mycobacteriosis, mediastinal tumor, sarcoidosis, pulmonary

**Table 4. Clinical stage and pathology of lung cancer detected by chest x-ray and low-dose CT screening.**

| | | Chest X-ray | Low-dose CT screening | | |
| --- | --- | --- | --- | --- | --- |
| | | | First examination | Second or later examinations | Total |
| Examinee | | 2218475* | 15,525 | 40,045 | 55,570 |
| Number of cases diagnosed with lung cancer | | 326* | 60 | 31 | 91 |
| Rate of detection of lung cancer (per 100,000 persons) | | 51.2** | 386.5 | 77.4 | 163.8 |
| 95% confidence interval | | 48.3–54.2 | 287.4–485.6 | 49.3–105.5 | 130.0–197.7 |
| Clinical stage | I | 72 | 59 | 31 | 90 |
| | II | 21 | 0 | 0 | 0 |
| | III | 32 | 1 | 0 | 1 |
| | IV | 32 | 0 | 0 | 0 |
| | Metastasis | 7 | 0 | 0 | 0 |
| | Unknown | 170 | 0 | 0 | 0 |
| | Total | 334*** | 60 | 31 | 91 |
| Pathology | Adenocarcinoma | 123 | 57 | 28 | 85 |
| | Squamous cell | 29 | 0 | 2 | 2 |
| | Small cell | 5 | 0 | 1 | 1 |
| | Large cell | 1 | 2 | 0 | 2 |
| | Metastasis | 7 | 0 | 0 | 0 |
| | Others | 0 | 1 | 0 | 1 |
| | Unknown | 169 | 0 | 0 | 0 |
| Total | | 334 | 60 | 31 | 91 |

* ≥40 years.

** Estimated rate (see Table 4).

*** All aged workers who were diagnosed with lung cancer.

fibrosis, aortic aneurysm, and interstitial pneumonia are presented. Cases of emphysema or nontuberculous mycobacteriosis were most often detected (rates, 0.40 [95% CI: 0.15–0.64] per 100,000 persons in men and 0.12 [95% CI: 0.07–0.18] in women).

## Discussion

This descriptive epidemiological survey reveals that the rate of TB detection among workers in Japan by mandatory annual CXR was estimated to be 1.8–5.1 per 100,000 persons. The number of new TB patients in the occupational workforce in 2016 was estimated to be 7.9 per 100,000 persons [4], and the expected morbidity rate from the age–sex distribution in this study population was 9.1 per 100,000 persons (including non-workers). From these results, the detection rate of TB by CXR was considered to be lower than the expected morbidity.

According to statistics from the Tuberculosis Surveillance Center of the Tuberculosis Research Institute, the rates of new cases identified from annual workplace health examinations, stratified by age, were reported to be 18.3%, 22.7%, 19.7%, 15.6%, and 6.0% for examinees in their 20s, 30s, 40s, 50s, and 60s, respectively [4]. Moreover, the latent period for TB has been reported to range from 1 or 2 months to 2 years, but most patients develop symptoms within 6 months [12]. Therefore, it is impossible to identify all instances of TB onset by annual CXR. Considering the national database statistics and the natural course of TB, it is difficult to appreciate the usefulness of routine annual CXR examination to detect TB in countries with low TB prevalence. Thus, the contribution of CXR to tuberculosis infection control might not be significant.

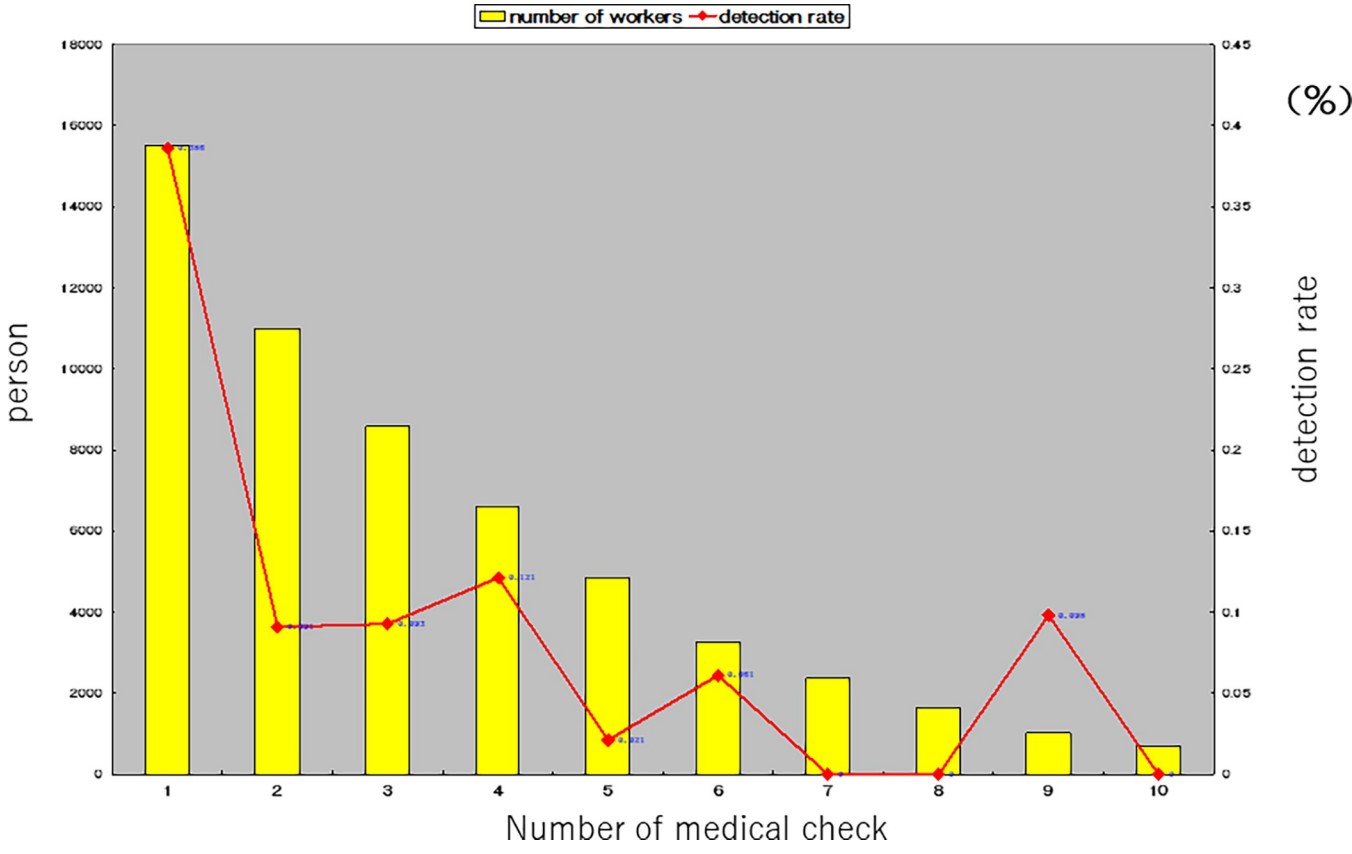

**Fig 1. Number of times that the examinees underwent screening with low-dose lung computed tomography scanning, and the rate of detection.**

The mass miniature radiophotography (MMR) sensitivity and specificity of CXR for the diagnosis of active TB were in the ranges of 64%–80% and 52%–63%, respectively, and thus, the investigation cannot be considered the gold standard for TB screening [13–16]. According to a recent study on the screening of immigrants, the sensitivity of CXR was reported to vary from 55.6% to 93% [17]. Our data showed that the positive predictive value of CXR for TB was 0.18%–0.56%, which was very low, given the low prevalence. The World Health Organisation aims to reduce the new TB cases by approximately 90% by 2035 and reduce the prevalence to <10 per 100,000 population [18]. From the perspective of cost-effectiveness, active case-findings for TB using CXR should be considered among high-risk populations. Nevertheless, the strategy should be re-considered when the prevalence is <10 per 100,000 persons in Japan [19]. Active case-findings among all general workers may be unnecessary.

Next, CXR plays a role in LC screening because CXR is recommended by governmental regulations as an alternative to LC screening [20]. Our data showed that the detection rate was 9.1–24.4 per 100,000 persons. This rate was lower, approximately 50%, than the expected rate of 41.0 per 100,000 persons of LC morbidity based on the age–sex distribution of this study population. However, CXRs conducted in the workplace differ from those performed for LC screening because CXR for cancer screening basically requires a comparative reading and a double reading [20,21]. In addition, sputum cytology must be performed for those with a smoking index of more than 600, but this has not been undertaken in annual health examinations in the workplace [21]. Furthermore, CXR in occupational areas are conducted annually, and were performed as an almost non-initial examination, which means that almost all

participants had been undergoing CXR every year. Thus, these different situations may be potentially underestimated. However, outcomes for the effect of cancer screening must be addressed by mortality, but not by increased numbers of diagnosis and more localized stages, because cancer screening has several biases and harmful effects such as overdiagnosis [22,23]. Regarding to LC screening, the main problem of CXR was reported to be a low sensitivity, while, that of LDCT was overdiagnosis [24]. Thus, using our data on this study, we cannot compare or evaluate the screening effects between CXR and LDCT. We only provided descriptive data as reference values. In Japan, Nawa et al. reported that LDCT among residents and workers had numerous advantages [10,11,25,26]. Recently, a population-based cohort study has been published by Hitachi City, Ibaraki Prefecture, and the LC mortality decreased by 51% in the group that was screened using LDCT than in the group screened using CXR [26].

Reports from two meta-analyses show that, compared with CXR, LDCT was more effective in reducing LC mortality [27,28]. In a randomized control trial conducted by the National Lung Screening Trial, the mortality in the LDCT group was reduced by 20% compared with that in the CXR group [29]. Furthermore, the European Union issued a statement to establish an LC screening system with LDCT [30].

Furthermore, challenges of LC screening using LDCT include the high cost and the high rate of false-positives due to the high sensitivity [31]. Therefore, previous reports have recommended that LDCT should be applied to high-risk populations, such as those over 50 years or smokers, to increase the prior probability [29,30]. At the beginning of LDCT screening in the Hitachi Health Care Center, a high detection rate of 384 per 100,000 persons was observed, despite the fact that CXRs were conducted yearly, which means that there were many cases of LC that could not be detected by CXR. In the 20 years of annual LDCT screening, the detection rate of LC has gradually decreased to 25.7 per 100,000 persons. This finding seems to indicate a limit of the annual detection rate for early LC, and is lower than the that (i.e., 155.2 per 100,000 persons) of expected morbidity, based on the age–sex distribution. This phenomenon is attributed to the characteristics of LDCT screening. Among many workers who annually undergo LDCT screening, suspected LC lesions were found in the first year. This is because LDCT is much more effective than CXR for detecting LC [10,11,25,26,32]. Most lesions found on LDCT were early LC (diameter, ≤20 mm) [10,11,25,26,32], and 6-mm nodules were detected [33]. Furthermore, a cohort study comparing the CT and CXR screening groups showed that LC deaths increased 5–7 years later in the CXR group than in the LDCT screening group [25]. Therefore, in the LDCT screening group, LC was detected more than 5–7 years earlier than in the CXR group [25,26]. Thus, as pointed out by the NLST, the morbidity and detection rates decreased in the subsequent yearly examination group.

In addition, the lower detection rate with LDCT than the estimated morbidity indicates one of the limitations of LDCT screening. When examining the histological type of LC death, it is assumed that workers with small cell carcinoma did not survive, although LC was detected using LDCT. This was because small-cell LC progresses quickly, and the number of cases that can be detected in early stages using LDCT may be limited [34]. Therefore, appropriate screening methods for small cell LC should be considered as a future measure.

CXR identified other diseases, such as emphysema, inflammatory changes, mediastinal tumors, pneumothorax, sarcoidosis, and aortic aneurysms. Moreover, the clinical utility should be considered separately from TB and LC. However, the incidence and/or prevalence rates of these diseases were very low. Thus, further investigations are needed to determine the cost-effectiveness of CXR for these diseases.

This study had several limitations. First, this was a descriptive survey. This is because all CXR in Japan have historically been started and performed as TB screening for workers. The number of detections as cancer screening cannot be evaluated due to problems of harmful

effects such as overdiagnosis. Thus, we could not effectively compare the usefulness of CXR and LDCT only based on our results. Second, only 38/88 institutions could follow-up the examinees who required follow-up examination. Moreover, only 30% of examinees were actually followed up. The estimated morbidity was described as a range, but accurate point estimation cannot be undertaken from our data. Third, the main results of LDCT were obtained from 1998 to 2008, and the results of the era might have been different for CXR. However, recent data from 2016 to 2018 did not differ considerably.

## Conclusions

In conclusion, based on the actual detection rate of TB using CXR, we ascertained that CXR plays a little role in TB detection in the general workplace. In addition, variability and reproducibility are well-known issues for CXR interpretation. If an outbreak of clustered transmission occurs in the workplace or if some LC cases are overlooked, retrospective re-evaluation of cases with a documented diagnosis who were previously evaluated using CXR may reveal discrepancies and create issues for the radiologist who interpreted the CXR. With regard to LC screening, the detection rate with CXR was lower, approximately 50%, than the expected rate of LC morbidity based on the age–sex distribution of this study population. However, the results of this study do not evaluate actual utility for CXR in the mandatory annual health examinations. Further follow-up studies are needed to assess mortality reduction in workplace for LC screening. Reports from North America have shown a significant reduction of mortality regarding LDCT screening, and it will be considered in high-risk populations, such as those over 50 years with heavy smokers in workplaces. Our study, however, demonstrated that the effectiveness of LDCT screening was limited for patients with small cell carcinoma. Thus, strengthening smoking cessation measures in workplaces should be considered as well as screening.

## Supporting information

**S1 Table. Expected morbidity of tuberculosis and lung cancer based on age–sex distribution of 42 institutions associated with NFHA\*.**
(DOCX)

**S2 Table. Results of recent low-dose CT examinations in 2016–2018.**
(DOCX)

**S3 Table. Characteristics of workers who died of lung cancer.**
(DOCX)

**S4 Table. Diseases other than tuberculosis and lung cancer detected by chest x-ray examination.**
(DOCX)

## Acknowledgments

We are grateful to Mr. Ichiji of National Federation of Industrial Health Organization for data collection. We also thank Mrs. A. Sakuyama for her secretarial assistance for this project.

## Author Contributions

**Conceptualization:** Yuya Watanabe, Toru Nakagawa, Takeshi Hayashi, Masayuki Tatemichi.

**Data curation:** Yuya Watanabe, Toru Nakagawa, Toru Honda, Takeshi Hayashi.

**Formal analysis:** Yuya Watanabe, Kota Fukai, Toru Honda, Masayuki Tatemichi.

**Funding acquisition:** Masayuki Tatemichi.

**Investigation:** Yuya Watanabe, Toru Nakagawa, Hiroyuki Furuya, Takeshi Hayashi, Masayuki Tatemichi.

**Project administration:** Yuya Watanabe, Toru Nakagawa, Kota Fukai, Toru Honda, Hiroyuki Furuya, Takeshi Hayashi, Masayuki Tatemichi.

**Resources:** Yuya Watanabe, Toru Nakagawa.

**Supervision:** Toru Nakagawa, Hiroyuki Furuya, Takeshi Hayashi, Masayuki Tatemichi.

**Writing – original draft:** Yuya Watanabe.

**Writing – review & editing:** Kota Fukai, Hiroyuki Furuya, Takeshi Hayashi, Masayuki Tatemichi.

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
