## [Decision Letter · Decision Letter 0]

17 May 2021

PONE-D-20-19580

Descriptive study of chest x-ray examination in mandatory annual health examinations at the workplace in Japan

PLOS ONE

Dear Dr. Tatemichi,

Thank you for submitting your manuscript to PLOS ONE. After careful consideration, we feel that it has merit but does not fully meet PLOS ONE’s publication criteria as it currently stands. Therefore, we invite you to submit a revised version of the manuscript that addresses the points raised during the review process.

The study seems well design and the manuscript well written. However, the reviewers do not totally agree with the interpretation of the data presented. Thus, I suggest major revision.

We look forward to receiving your revised manuscript.

Kind regards,

Alessandra Giuliani

Academic Editor

PLOS ONE

Additional Editor Comments:

The study seems well design and the manuscript well written. However, the reviewers do not totally agree with the interpretation of the data presented. Thus, I suggest major revision.

Journal Requirements:

[This study was supported by the Ministry of Health, Labour, and Welfare of Japan through an Industrial Disease Clinical Research Grants (grant no. 170301).].    

We note that one or more of the authors are employed by a commercial company: Hitachi Health Care Center

Reviewers' comments:

Reviewer's Responses to Questions

**Comments to the Author**

1. Is the manuscript technically sound, and do the data support the conclusions?

Reviewer #1: Partly

Reviewer #2: No

2. Has the statistical analysis been performed appropriately and rigorously? 

Reviewer #1: Yes

Reviewer #2: Yes

3. Have the authors made all data underlying the findings in their manuscript fully available?

Reviewer #1: No

Reviewer #2: Yes

4. Is the manuscript presented in an intelligible fashion and written in standard English?

Reviewer #1: Yes

Reviewer #2: Yes

5. Review Comments to the Author

Reviewer #1: Dr. Watanabe analyzed the national occupational health examination database to appreciate the effectiveness of routine annual chest x-ray in Japan. The main findings were (1) Among the 4764985 participants who underwent chest x-ray examination for TB detection and follow-up data for diagnosis were available, 47962 subjects were advised for follow up, 16734 subjects underwent follow-up exam, and 88 patients were diagnosed of active TB (rate of detection 1.8/100000, positive predictive value 0.53%); (2) Among 3688395 participants who underwent chest x-ray for lung cancer detection, 38146 subjects were advised for follow-up examination, 14137 subjects underwent follow-up exam, and 334 cases were diagnosed with lung cancer (rate of detection 9.1/100000, positive predictive value 2.36%). (3) Stage I lung cancer accounted for 43.9% among the lung cancer revealed by annual chest x-ray examination. The authors concluded that chest x-ray examination plays a role in the detection of active TB and early stage lung cancer.

The study is well design and the manuscript is well written. However, I do not totally agree with the interpretation of the data presented. I would suggest some minor revisions.

1. I am confused with the calculation of positive predictive value for TB detection and lung cancer. For the positive predictive value of chest x-ray to detect active TB, the authors use the number of cases diagnosed with tuberculosis (88) as the numerator and the number of participants who underwent follow-up (16734) as the denominator. Were all participants advised for follow-up examination due to suspicion of active TB? If yes, then the positive predictive value should be 88/47962 = 1.8%. If no, then the denominator should be the number of subjects whose chest x-ray were considered possible active TB. Similar issues existed in the calculation of positive predictive value of lung cancer detection.

2. According to the estimation by the authors (line 172 of the manuscript), the expected incidence of active TB for the population studied was 9.1/100000. However, the rate of detection was 1.8/100000. That means more than 80% of the cases of active were not detected by annual chest x-ray examination. With an extremely low positive predictive value (0.53%), it is difficult to appreciate the usefulness of routine annual chest x-ray examination for the purpose of TB detection in countries with low prevalence of TB. The variability and reproducibility were well-known issues for chest x-ray interpretation. If an outbreak of clustered transmission among work place occurs, retrospective re-evaluation of the chest x-ray with a documented diagnosis in mind may reveal some obscured abnormalities and cause trouble to the radiologist who firsthand interpreted the chest x-ray. Similar issues existed in the interpretation of usefulness of lung cancer detection with chest x-ray. At least, these concerns should be listed.

3. The expected incidence of lung cancer increases exponentially in the elderly. I suggest the authors to extend the table 3 by further stratifying the detection of lung cancer among age groups of <40, 40-60, and >60 years old.

4. The table 5 is not quite relevant to the topics of the present study. It is difficult to determine the accuracy, sensitivity, and specificity by chest x-ray examination for the diagnoses listed. I would suggest to remove the information of table 5 from the present report.

5. In line 256-258, the sentence “From the perspective of cost-effectiveness, screening for Tb should be considered when the prevalence is lower than 10 per 100,000 persons” is ambiguous.

Reviewer #2: Numbers of diagnoses and stage are poor surrogate outcomes for the effect of health checks. Increased numbers of diagnosis and more localised stages can be and impact of a beneficial effect, a harmful effect (especially overdiagnosis) or both. Moreover, mortality rates in a non-randomised trial cannot be used as a result of benefits. Finally, survival rates will be biased due to lead time bias and leangt time bias (overdiagnosis bias) and are the not useful outcomes when it comes to early disease detection, e.g. screening and health checks.

6. PLOS authors have the option to publish the peer review history of their article (what does this mean?). If published, this will include your full peer review and any attached files.

Reviewer #1: No

Reviewer #2: **Yes: **John Brodersen

---

## [Author Response · Author response to Decision Letter 0]

19 Aug 2021

Reviewer #1:

 Dr. Watanabe analyzed the national occupational health examination database to appreciate the effectiveness of routine annual chest x-ray in Japan. The main findings were (1) Among the 4764985 participants who underwent chest x-ray examination for TB detection and follow-up data for diagnosis were available, 47962 subjects were advised for follow up, 16734 subjects underwent follow-up exam, and 88 patients were diagnosed of active TB (rate of detection 1.8/100000, positive predictive value 0.53%); (2) Among 3688395 participants who underwent chest x-ray for lung cancer detection, 38146 subjects were advised for follow-up examination, 14137 subjects underwent follow-up exam, and 334 cases were diagnosed with lung cancer (rate of detection 9.1/100000, positive predictive value 2.36%). (3) Stage I lung cancer accounted for 43.9% among the lung cancer revealed by annual chest x-ray examination. The authors concluded that chest x-ray examination plays a role in the detection of active TB and early stage lung cancer.

The study is well design and the manuscript is well written. However, I do not totally agree with the interpretation of the data presented. I would suggest some minor revisions.

Response: The authors would like to thank the reviewer for the feedback. We have made every effort to address the issues raised and to respond to all comments. The revisions are indicated in red color text in the revised manuscript. In addition, we have carried our English check by the native speakers again. Please, find next a detailed, point-by-point response to the comments below.

Review１

1. I am confused with the calculation of positive predictive value for TB detection and lung cancer. For the positive predictive value of chest x-ray to detect active TB, the authors use the number of cases diagnosed with tuberculosis (88) as the numerator and the number of participants who underwent follow-up (16734) as the denominator. Were all participants advised for follow-up examination due to suspicion of active TB? If yes, then the positive predictive value should be 88/47962 = 1.8%. If no, then the denominator should be the number of subjects whose chest x-ray were considered possible active TB. Similar issues existed in the calculation of positive predictive value of lung cancer detection.

Response: As pointed out by the reviewer, active TB was detected in 88 patients. Thus, the PPV was actually 0.18% (88/47962). However, we could follow up only 16734 (34.9%) participants who underwent follow-up examination. The measured PPV of 0.18% was the lowest PPV and might have been underestimated. The maximum PPV was calculated as 0.56% (88/16734), which might have been overestimated. Thus, plausible PPV was estimated to range between 0.18% and 0.56%.

We have provided this information in the revised manuscript as follows: 

‘The positive predictive value (PPV) was primarily calculated to be 0.18% (88/47,962). This value might have been an underestimation, because the cases that could be followed up were 16,734 (34.9%). Thus, the maximum PPV was calculated as 0.56% (88/16734), which might have been an overestimation. Plausible PPV was estimated to range between 0.18% and 0.56%.’ (Lines 171–175)

Moreover, we have revised the data presented in Tables 2 and 3.

2. According to the estimation by the authors (line 172 of the manuscript), the expected incidence of active TB for the population studied was 9.1/100000. However, the rate of detection was 1.8/100000. That means more than 80% of the cases of active were not detected by annual chest x-ray examination. With an extremely low positive predictive value (0.53%), it is difficult to appreciate the usefulness of routine annual chest x-ray examination for the purpose of TB detection in countries with low prevalence of TB. The variability and reproducibility were well-known issues for chest x-ray interpretation. If an outbreak of clustered transmission among work place occurs, retrospective re-evaluation of the chest x-ray with a documented diagnosis in mind may reveal some obscured abnormalities and cause trouble to the radiologist who firsthand interpreted the chest x-ray. Similar issues existed in the interpretation of usefulness of lung cancer detection with chest x-ray. At least, these concerns should be listed.

Response:

We would like to thank the reviewer for the comment. Indeed, the estimated prevalence of tuberculosis was 9.1/100,000 persons, and the detection rate on chest X-rays was 1.8/100,000 persons. The estimated prevalence of lung cancer was 41.0/100,000 persons, while the detection rate was 9.1/100,000 persons. Since the detection rate of chest X-rays is considered to be limited, we have revised the corresponding part in the Discussion section as follows: 

(Before editing)

Line 247-250: Considering the statistics from the national database and the natural course of TB, CXR examination in periodic workplace health examinations could be expected to actually play a role to some extent in the detection of active TB in clinical practice. However, the contribution of CXR to tuberculosis infection control might not be deemed essential.

Line 329-330: In conclusion, based on the actual rate of detection of TB using CXR, we ascertained that CXR plays a certain role in TB detection at the workplace. 

(After editing)

‘Considering the national database statistics and the natural course of TB, it is difficult to appreciate the usefulness of routine annual CXR examination to detect TB in countries with low TB prevalence. Thus, the contribution of CXR to tuberculosis infection control might not be significant.’. (Lines 262–265)

‘In conclusion, based on the actual detection rate of TB using CXR, we ascertained that CXR plays a little role in TB detection in the general workplace. In addition, variability and reproducibility are well-known issues for CXR interpretation. If an outbreak of clustered transmission occurs in the workplace or if some LC cases are overlooked, retrospective re-evaluation of cases with a documented diagnosis who were previously evaluated using CXR may reveal discrepancies and create issues for the radiologist who interpreted the CXR’. (Lines 348–353)

3. The expected incidence of lung cancer increases exponentially in the elderly. I suggest the authors to extend the table 3 by further stratifying the detection of lung cancer among age groups of <40, 40-60, and >60 years old.

Response: 

Please note that those who underwent CXR examination were aged ≤60 years, as the retirement age was at 60 years in 2016. Thus, the number of those aged >60 years was negligible.

4. The table 5 is not quite relevant to the topics of the present study. It is difficult to determine the accuracy, sensitivity, and specificity by chest x-ray examination for the diagnoses listed. I would suggest to remove the information of table 5 from the present report.

Response:

We would like to thank the reviewer for the constructive comment. According to the reviewer’s suggestions, we have revised our manuscript and removed the information presented in Table 5.

Examination of chest X-rays at workplace aimed to screen tuberculosis and lung cancer. Thus, this study included descriptive information regarding the observed stage of disease according to CXR findings. Therefore, we have moved this Table to the Supplementary files.

5. In line 256-258, the sentence “From the perspective of cost-effectiveness, screening for Tb should be considered when the prevalence is lower than 10 per 100,000 persons” is ambiguous.

Response: 

We would like to thank the reviewer for the comment. Please note that we have revised this part as follows: ‘The World Health Organisation aims to reduce the new TB cases by approximately 90% by 2035 and reduce the prevalence to <10 per 100,000 population [19]. From the perspective of cost-effectiveness, active case-findings for TB using CXR should be considered among high-risk populations. Nevertheless, the strategy should be re-considered when the prevalence is <10 per 100,000 persons in Japan [20]. Active case-findings among all general workers may be unnecessary.’. (Lines 271–277)

Reviewer #2: 

Numbers of diagnoses and stage are poor surrogate outcomes for the effect of health checks. Increased numbers of diagnosis and more localised stages can be and impact of a beneficial effect, a harmful effect (especially overdiagnosis) or both. Moreover, mortality rates in a non-randomised trial cannot be used as a result of benefits. Finally, survival rates will be biased due to lead time bias and leangt time bias (overdiagnosis bias) and are the not useful outcomes when it comes to early disease detection, e.g. screening and health checks.

Response:

The authors would like to thank the reviewer for the constructive feedback and suggestions to improve the manuscript. We have made every effort to address the issues raised and to respond to the reviewer’s comments. The revisions are indicated in red color text in the revised manuscript. In addition, we have carried our English check by the native speakers again.

We fully understood that the evaluation of screening such as cancer screening should be based on the first priority demonstrating its effectiveness as screening by randomized controlled trials. Many studies have already conducted studies on the effectiveness of chest X-rays on lung cancer screening. However, it is only an evaluation of the effectiveness of a screening method by experimental research protocols with limited sample size. Thus, to evaluate usefulness as a measure, a realistic evaluation when implemented in society is also necessary. The purpose of this study is to provide actual descriptive data on the current state of social implementation of the screening as a measure, not to evaluate the effectiveness of the screening. It only shows social significance.

Please note that we have added the following part to the Introduction section of the revised manuscript to address the issues raised by the reviewer and to clearly describe our purpose:

‘The effectiveness for screening diseases such as cancer should be determined with randomised controlled trials using mortality as the outcome. The Japanese guidelines for lung cancer screening recommend the performance of CXR [9]. However, these guidelines were developed based on the findings of experimental studies with a limited sample size that evaluated the effectiveness of screening methods. Thus, to evaluate significance as a measure, a realistic evaluation when implemented in a large population is also needed. The purpose of this study was to provide actual descriptive data on the current state of social implementation of screening, and not to evaluate the effectiveness of the screening method.’. (Lines 67–74)

---

## [Decision Letter · Decision Letter 1]

27 Oct 2021

PONE-D-20-19580R1Descriptive study of chest x-ray examination in mandatory annual health examinations at the workplace in JapanPLOS ONE

Dear Dr. Tatemichi,

Thank you for submitting your manuscript to PLOS ONE. After careful consideration, we feel that it has merit but does not fully meet PLOS ONE’s publication criteria as it currently stands. Therefore, we invite you to submit a revised version of the manuscript that addresses the points raised during the review process.

 Since Reviewer 2 was not adequately satisfied with your previous review, I refer you to request a further review by asking you to focus on the responses to Reviewer 2.

We look forward to receiving your revised manuscript.

Kind regards,

Alessandra Giuliani

Academic Editor

PLOS ONE

Additional Editor Comments (if provided):

Dear authors,

Since Reviewer 2 was not adequately satisfied with your previous review, I refer you to request a further review by asking you to focus on the responses to Reviewer 2.

Reviewers' comments:

Reviewer's Responses to Questions

**Comments to the Author**

1. If the authors have adequately addressed your comments raised in a previous round of review and you feel that this manuscript is now acceptable for publication, you may indicate that here to bypass the “Comments to the Author” section, enter your conflict of interest statement in the “Confidential to Editor” section, and submit your "Accept" recommendation.

Reviewer #1: All comments have been addressed

Reviewer #2: (No Response)

2. Is the manuscript technically sound, and do the data support the conclusions?

Reviewer #1: Yes

Reviewer #2: No

3. Has the statistical analysis been performed appropriately and rigorously? 

Reviewer #1: Yes

Reviewer #2: N/A

4. Have the authors made all data underlying the findings in their manuscript fully available?

Reviewer #1: Yes

Reviewer #2: Yes

5. Is the manuscript presented in an intelligible fashion and written in standard English?

Reviewer #1: Yes

Reviewer #2: Yes

6. Review Comments to the Author

Reviewer #1: Thank you. All comments were addressed appropriately. I believe this revision is appropriate to be accepted for publication.

Reviewer #2: The authors have not responded adequately to my previous critique and have not changed their interpretation of the outcomes of numbers of diagnoses and stage, e.g. the authors have not discussed the possibility of overdiagnosis at all. Their conclusion is also unchanged and is most likely not right.

7. PLOS authors have the option to publish the peer review history of their article (what does this mean?). If published, this will include your full peer review and any attached files.

Reviewer #1: **Yes: **Chih-Hsin Lee

Reviewer #2: **Yes: **John Brodersen

---

## [Author Response · Author response to Decision Letter 1]

25 Nov 2021

The authors would like to thank the reviewer for the constructive feedback and suggestions to improve the manuscript. We have made every effort to address the issues raised and to respond to the reviewer’s comments. The revisions are indicated in red font in the revised manuscript. 

Please note that we have revised the following part to the Abstract, Introduction, and Discussion section of the revised manuscript to address the issues raised by the reviewer and to describe our purpose clearly:

As the reviewer mentioned, the numbers of the detection and early-stage cancer are generally not useful outcomes, mainly because they can include cases with overdiagnosis. Particularly the problem of LDCT-based lung cancer screening is overdiagnosis. Thus, the effectiveness for screening diseases such as cancer must be determined with randomised controlled trials using mortality as the outcome. Lung cancer screening with CXR is conducted based on evidence of the effect on reduction of mortality. However, in CXR, the problem of sensitivity is greater than the problem of overdiagnosis in terms of screening efficacy. The purpose of this study was to provide actual descriptive data on the current state of social implementation of screening among workers and not to evaluate the effectiveness of the screening method.

Our study investigated the detection rate and early cancer detection rate in medical examinations that have already been implemented as a law in the workplace for public health.

Therefore, we revised our interpretation and conclusion according to reviewer’s comments.

This revision appears in red font.

Abstract

Before:

L17-18:

This study aimed to assess the clinical utility of chest X-ray examination in the workplace and provide basic data to consider future policies for mandatory annual health examinations.

Revised:

L17-18: 

This study aimed to provide basic data to consider future policies for mandatory annual health examinations in the workplace.

Before;

L36-38

Early-stage lung cancer can also be detected among workers by Chest x-ray examinations. However, low-dose computed tomography may be superior for lung cancer screening because it can detect twice as many cases as chest radiography.

Revised:

L34-38:

With regard to LC screening, the detection rate of LC was lower, approximately 50%, than the expected rate (41.0 per 100,000 persons) of LC morbidity based on the age-sex distribution of this study population. However, the role of CXR for LC screening cannot be mentioned based on this result because assessment of mortality reduction is essential to evaluate.

Introduction

Before:

L66-73

The effectiveness for screening diseases such as cancer should be determined with randomised controlled trials using mortality as the outcome. The Japanese guidelines for lung cancer screening recommend the performance of CXR [9]. However, these guidelines were developed based on the findings of experimental studies with a limited sample size that evaluated the effectiveness of screening methods. Thus, to evaluate significance as a measure, a realistic evaluation when implemented in a large population is also needed. The purpose of this study was to provide actual descriptive data on the current state of social implementation of screening, and not to evaluate the effectiveness of the screening method.

Revised:

L66-74

Japanese guideline for lung cancer screening recommends CXR [9]. Thus mandatory CXR in workplace plays a role in lung cancer screening in addition to screening for TB. The first priority of evaluation of effectiveness on cancer screening must be based on randomized controlled trials (RCT) using mortality as an outcome. Theoretically, the detection number of cancers is not a useful outcome to evaluate effectiveness because cancers detected by screening can include cases with overdiagnosis. Thus this study did not aim to evaluate the effectiveness of CXR for cancer screening. The purpose of this study is to provide actual descriptive data on the current state of social implementation of the screening as a measure, not directly to assess the effectiveness of CXR for cancer screening.

Discussion

Before:

L293-294

Merely by considering this result, the early-stage detection rate with LDCT was two times higher than that by CXR.

Revised:

L287-293

However, outcomes for the effect of cancer screening must be addressed by mortality, but not by increased numbers of diagnoses and more localized stages, because cancer screening has several biases and harmful effects such as overdiagnosis [23,24]. Regarding LC screening, the main problem of CXR examination was reported to be a low sensitivity while that of LDCT was overdiagnosis [25]. Thus, using our data on this study, we cannot compare or evaluate the screening effects between CXR and LDCT. We only provided descriptive data as reference values. 

References

Three references have been added.

23. Brawley OW, Kramer BS. Cancer screening in theory and in practice. J Clin Oncol 2005;23:293-300.

24. Raffle A, Gray M. Screening: evidence and practice. Oxford University Press, 2007.

25. T Sobue 1, T Suzuki, M Matsuda, T Horai, A Kajita, K Kuriyama, M Fukuoka, Y Kusunoki, M Kikui, S Ryu, et al. Sensitivity and specificity of lung cancer screening in Osaka, Japan. Jpn J Cancer Res. 1991;82(10):1069-76.

Before:

L298-299

‘’Moreover, this finding indicated that the utility of LDCT for screening for LC was almost twice as high as that of CXR.’’ 

Revised:

This sentence has been deleted.

Before:

L337-338

 We could not effectively compare the usefulness of CXR and LDCT.

Revised:

L335-339

This is because all CXR in Japan has historically been started and performed as TB screening for workers. The number of detections as cancer screening cannot be evaluated due to problems of harmful effects such as overdiagnosis. Thus, we could not effectively compare the usefulness of CXR and LDCT only based on our results.

Before:

L351-358

With regard to LC screening, the detection rate with CXR was lower than that with LDCT, and the detection rate in Stage I using CXR, when detection may save lives, was approximately 50% lower than that using LDCT. These findings were consistent with those of a previous residence-based comparative study by Nawa et al [23]. Nevertheless, our results do not support the recommendation for the exclusion of CXR from the mandatory annual health examinations; however, effective introduction of LDCT screening to workplace health examinations should be considered as a substitute for CXR for LC screening in the future.

Revised:

L352-359

With regard to LC screening, the detection rate with CXR was lower, approximately 50%, than the expected rate of LC morbidity based on the age-sex distribution of this study population. However, the results of this study do not evaluate the actual utility of CXR in the mandatory annual health examinations. Further follow-up studies are needed to assess mortality reduction in workplace for LC screening. Inductions of LDCT are discussed for LC screening worldwide, the effective introduction of LDCT screening to workplace health examinations should be considered in the future, based on the evidence by RCT in Japan.

---

## [Decision Letter · Decision Letter 2]

14 Dec 2021

PONE-D-20-19580R2Descriptive study of chest x-ray examination in mandatory annual health examinations at the workplace in JapanPLOS ONE

Dear Dr. Tatemichi,

Thank you for submitting your manuscript to PLOS ONE. After careful consideration, we feel that it has merit but does not fully meet PLOS ONE’s publication criteria as it currently stands. Therefore, we invite you to submit a revised version of the manuscript that addresses the points raised during the review process.

In the Conclusion section of this revised version. The sentence is ambiguous: "Inductions of LDCT are discussing for LC screening worldwidely...". Please revise it for clarity.

We look forward to receiving your revised manuscript.

Kind regards,

Alessandra Giuliani

Academic Editor

PLOS ONE

Journal Requirements:

Additional Editor Comments (if provided):

In the Conclusion section of this revised version. The sentence is ambiguous: "Inductions of LDCT are discussing for LC screening worldwidely...". Please revise it for clarity.

Reviewers' comments:

Reviewer's Responses to Questions

**Comments to the Author**

1. If the authors have adequately addressed your comments raised in a previous round of review and you feel that this manuscript is now acceptable for publication, you may indicate that here to bypass the “Comments to the Author” section, enter your conflict of interest statement in the “Confidential to Editor” section, and submit your "Accept" recommendation.

Reviewer #1: All comments have been addressed

2. Is the manuscript technically sound, and do the data support the conclusions?

Reviewer #1: Yes

3. Has the statistical analysis been performed appropriately and rigorously? 

Reviewer #1: Yes

4. Have the authors made all data underlying the findings in their manuscript fully available?

Reviewer #1: Yes

5. Is the manuscript presented in an intelligible fashion and written in standard English?

Reviewer #1: Yes

6. Review Comments to the Author

Reviewer #1: In the Conclusion section of this revised version. The sentence is ambiguous: "Inductions of LDCT are discussing for LC screening worldwidely...". Please revise it for clarity.

7. PLOS authors have the option to publish the peer review history of their article (what does this mean?). If published, this will include your full peer review and any attached files.

Reviewer #1: **Yes: **Chih-Hsin Lee

---

## [Author Response · Author response to Decision Letter 2]

20 Dec 2021

Reviewer #1: 

In the Conclusion section of this revised version. The sentence is ambiguous: "Inductions of LDCT are discussing for LC screening worldwidely...". Please revise it for clarity.

Response:

We would like to appreciate the Reviewers and the Editorial Board for taking the time to review our manuscript and thank their constructive and thoughtful comments. According to the suggestion, we have revised our manuscript below.

Conclusion

Before:

L357-359:

Inductions of LDCT are discussing for LC screening worldwidely, effective introduction of LDCT screening to workplace health examinations should be considered in the future, based on the evidence by RCT in Japan.

Revised:

L357-362: 

Reports from North America have shown a significant reduction of mortality regarding LDCT screening, and it will be considered in high-risk populations, such as those over 50 years with heavy smokers in workplaces. Our study, however, demonstrated that the effectiveness of LDCT screening was limited for patients with small cell carcinoma. Thus, strengthening smoking cessation measures in workplaces should be considered as well as screening.

---

## [Editor Report · Decision Letter 3]

27 Dec 2021

Descriptive study of chest x-ray examination in mandatory annual health examinations at the workplace in Japan

PONE-D-20-19580R3

Dear Dr. Tatemichi,

We’re pleased to inform you that your manuscript has been judged scientifically suitable for publication and will be formally accepted for publication once it meets all outstanding technical requirements.

Kind regards,

Alessandra Giuliani

Academic Editor

PLOS ONE
---

## [Editor Report · Acceptance letter]

4 Jan 2022

PONE-D-20-19580R3 

Descriptive study of chest x-ray examination in mandatory annual health examinations at the workplace in Japan 

Dear Dr. Tatemichi:

I'm pleased to inform you that your manuscript has been deemed suitable for publication in PLOS ONE. Congratulations! Your manuscript is now with our production department. 

Kind regards, 

on behalf of

Dr. Alessandra Giuliani 

Academic Editor

PLOS ONE